# Neuroprotective Effect of Dioscin against Parkinson’s Disease via Adjusting Dual-Specificity Phosphatase 6 (DUSP6)-Mediated Oxidative Stress

**DOI:** 10.3390/molecules27103151

**Published:** 2022-05-14

**Authors:** Zhang Mao, Meng Gao, Xuerong Zhao, Lili Li, Jinyong Peng

**Affiliations:** 1College of Pharmacy, Dalian Medical University, Western 9 Lvshunnan Road, Dalian 116044, China; dy2020maozhang@126.com (Z.M.); dy2020gaomeng@126.com (M.G.); dylilinxia2020@163.com (X.Z.); 2College of Intergrative Medicine, Dalian Medical University, Western 9 Lvshunnan Road, Dalian 116044, China; 3Department of Pharmacy, Anhui University of Chinese Medicine, Hefei 230012, China

**Keywords:** Parkinson’s disease, dioscin, oxidative stress, DUSP6 signal

## Abstract

Exploration of lead compounds against Parkinson’s disease (PD), a neurodegenerative disease, is of great important. Dioscin, a bioactive natural product, shows various pharmacological effects. However, the activities and mechanisms of dioscin against PD have not been well investigated. In this study, the tests on 6-hydroxydopamine (6-OHDA)-induced PC12 cells and rats were carried out. The results showed that dioscin dramatically improved cell viability, decreased reactive oxygen species (ROS) levels, improved motor behavior and tyrosine hydroxylase(TH) levels and restored the levels of glutathione (GSH) and malondialdehyde (MDA) in rats. Mechanism investigation showed that dioscin not only markedly increased the expression level of dual- specificity phosphatase 6 (DUSP6) by 1.87-fold in cells and 2.56-fold in rats, and decreased phospho-extracellular regulated protein kinases (p-ERK) level by 2.12-fold in cells and 2.34-fold in rats, but also increased the levels of nuclear factor erythroid2-related factor 2 (Nrf2), hemeoxygenase-1 (HO-1), superoxide dismutase (SOD) and decreased the levels of kelch-1ike ECH-associated protein l (Keap1) in vitro and in vivo. Furthermore, DUSP6 siRNA transfection experiment in PC12 cells validated the protective effects of dioscin against PD via regulating DUSP6 to adjust the Keap1/Nrf2 pathway. Our data supported that dioscin has protection against PD in regulating oxidative stress via DUSP6 signal, which should be considered as an efficient candidate for the treatment of PD in the future.

## 1. Introduction

Parkinson’s disease (PD) is one of the most prevalent neurodegenerative diseases worldwide which targets people aged 55 or over. The causes of substantia nigra striatum lesions are mainly related to genetics, environmental factors, mitochondrial dysfunction, immune abnormalities, neuroinflammation and oxidative stress [1,2]. Mitochondrial DNA is vulnerable to the attack of free radicals and reactive oxygen species (ROS) in mitochondria, which is more likely to occur in the high oxidative stress environment of dopaminergic cells in substantia nigra [3]. Thus, antioxidant therapy opens up a new direction to treat PD.

Dual-specificity phosphatases (DUSPs) have been acknowledged as important regulators of signaling pathways to affect various physiological processes [4]. Dual -specificity phosphatases 6 (DUSP6), a member of a subfamily of protein tyrosine phosphatases, can dephosphorylate extracellular signal-regulated protein kinase 1/2 (ERK1/2) to adjust the signal and affect cell proliferation, differentiation and apoptosis [5]. As a short-lived protein, DUSP6 has been reported that some oxidation factors can cause DUSP6 degradation and up-regulate ERK1/2 phosphorylation. Simultaneously, DUSP6 can protect against ischemic brain damage in an oxygen and glucose deprivation /reoxygenation model [6]. Additionally, a MKP3/DUSP6-dependent signal can inhibit oxidative stress-induced ERK phosphorylation and neurotoxicity in SH-SY5Y cells [7]. However, the effects and mechanism of DUSP6 involvement in oxidative stress on PD have not been reported.

Dioscin (Figure 1), a natural steroidal saponin, exists in some medicinal plants, which is also an active component in common food Shanyao [8]. Dioscin has excellent activities on metabolism, anti-cancer, anti-inflammatory and oxidative stress [9,10]. Our previous studies displayed that dioscin can attenuate cerebral ischemia/reperfusion injury [11], and it has a neuroprotective effect against brain aging via decreasing oxidative stress [12]. Dioscin–zein–carboxymethyl cellulose (CMC) complex can prevent PD in Caenorhabditis elegans [13]. However, the effects and mechanisms of dioscin against PD have not been well reported.

Therefore, we hypothesized that dioscin could ameliorate oxidative stress and brain injury against PD through adjusting the DUSP6/ERK pathway. These results provided one potent candidate for preventing and treating PD.

## 2. Results

### 2.1. Dioscin Alleviated 6-OHDA-Mediated Injury in PC12 Cells

6-OHDA, a hydroxylated analog of dopamine, is taken up through dopamine transporters and can cause cytotoxicity to PC12 cells which is used for cell modeling in our work. As shown in Figure 2A, dioscin at the concentrations from 0 to 580 nM exerted no toxicity to PC12 cells for 6, 12 and 24 h. We next tested 6-OHDA-induced (0, 25, 50, 100, 200 and 400 μM) toxicity on PC12 cells. As shown in Figure 2B, 6-OHDA in the range of 50–400 μM significantly decreased cell viability and cell death was decreased to 54.32% by 400 μM of 6-OHDA for 24 h. Hence, 6-OHDA at the concentration of 400 μM for 24 h treatment was selected to induce cell injury. As shown in Figure 2C, compared with 6-OHDA group, dioscin at the concentrations of 145, 290 and 580 nM for 12 h restored cell viability by 1.06-, 1.59- and 1.58-fold, respectively. Hence, dioscin under 145, 290 and 580 nM for 12 h treatment was selected for the following tests. In addition, cell morphological changes were found by bright field images after dioscin treatment (Figure 2D).

### 2.2. Dioscin Decreased Oxidative Stress in PC12 Cells

As shown in Figure 3A,B, compared with the 6-OHDA group, dioscin markedly alleviated the ROS level in PC12 cells. LDH activities in dioscin high-dose, middle-dose and low-dose groups were significantly decreased by 1.46-, 1.77- and 2.07-fold. In addition, as shown in Figure 3C, the decreased GSH level and increased MDA level were dramatically reversed in dioscin-treated groups compared with the model group.

### 2.3. Dioscin Improved Motor Behavior In Vivo

As shown in Figure 4A, the animals in the control group traversed the walkway with short consistent steps, while the rats in the 6-OHDA group walked at a slower irregular pace with fewer steps per second. After dioscin treatment, the long uneven steps were decreased to consistent repetitive short steps. As shown in Figure 4B, the stand phase, step cycle, run variation and base of support revealed that the period of contact between all paws and the glass plate were increased in 6-OHDA group, which were decreased by dioscin. In addition, the stride length was significantly improved after dioscin administration compared with the 6-OHDA group. Apomorphine-induced rotation test in Figure 4C showed that the apomorphine-induced rotation was observed in the 6-OHDA group, which was alleviated by dioscin. Taken together, the results on a dose- dependent improvement of gait parameters induced by 6-OHDA were protected by dioscin.

### 2.4. Dioscin Activated TH Expression In Vivo

To further evaluate the neuroprotective effect of dioscinin vivo, H&E staining, TH expression andthe levels ofGSH and MDA were tested. H&E results in Figure 5A showed that the brain tissue structure of control rats was clear and complete. Cell necrosis and tissue structure damage were observed in the 6-OHDA group. After dioscin treatment, the degree of injury was alleviated. Down-regulated TH expression suggested the inhibition of dopamine neurons in the 6-OHDA group compared with the control group, which was markedly reversed after dioscin administration. In addition, the results in Figure 5B also showed the same conclusion. The above results suggested that dioscin exerted neuroprotective effects by increasing TH expression. As shown in Figure 5C, the GSH level was markedly reversed by 1.08-, 1.89- and 2.32-fold, and the MDA level was decreased by 1.34-, 1.46- and1.93-fold by dioscin. This result indicated that dioscin exhibiteda protective effect against 6-OHDA-induced TH down-regulation and oxidative damage.

### 2.5. Dioscin Activated DUSP6 Expression and ERK Phosphorylation

As shown in Figure 6A, the results of in vitro and in vivo immunofluorescence staining revealed that the expression levels of DUSP6 in the model groups were significantly reduced, which were markedly increased by dioscin. As shown in Figure 6B, compared with the model group, the expression level of DUSP6 in PC12 cells was up- regulated by 1.87-fold in the dioscin high-dose group, and the expression levels of DUSP6 in rats were up-regulated by 2.34-fold and 2.56-fold in the dioscin middle-dose and high-dose groups. Compared with the 6-OHDA group, the levels of p-ERK in the dioscin high-dose group were down-regulated by 2.12-fold in cells and 2.34-foldin rats. These results suggested that dioscin suppressed 6-OHDA-induced cytotoxicity via the activation of the DUSP6/ERK pathway.

### 2.6. Dioscin Adjusted the Keap1/Nrf2 Signal Pathway

The results in Figure 7A,B revealed that in the 6-OHDA group, the expression levels of Keap1 were significantly up-regulated by 2.23-fold in cells and 2.98-fold in rats, and the expression levels of Nrf2, HO-1 and SOD were markedly decreased by 2.12-fold, 2.45-fold and 2.65-fold in cells, and 1.66-fold, 1.98-fold and 2.32-fold in rats, compared with the control group. These results were dramatically reversed by dioscin in vitro and in vivo. The findings indicated that the protective effect of dioscin against PD should be through regulating Keap1/Nrf2 signaling.

### 2.7. Effects of Dioscin after the Blockade of DUSP6

The results in Figure 8A showed that dioscin increased cell viability compared with the 6-OHDA group, while DUSP6 siRNA transfection reversed the effect. As shown in Figure 8B,C, dioscin up-regulated the expression level of DUSP6 and decreased the p-ERK level, and DUSP6 siRNA transfection restrained the expression levels of DUSP6 and p-ERK. As shown in Figure 9A, dioscin markedly alleviated the ROS level compared with the 6-OHDA group, which was down-regulated in the dioscin-treated groupafter transfection. As shown in Figure 9B, dioscin clearly increased the GSH level and decreased the levels of MDA and LDH with or without transfection of DUSP6 siRNA. As shown in Figure 9C, the expression of Keap1 was down-regulate, and the expression levels of Nrf2, HO-1 and SOD were markedly increased in the dioscin-treated group after transfection. These results indicated that dioscin attenuated 6-OHDA-induced PD injury via DUSP6-mediated oxidative stress.

## 3. Discussion

PD, a complex neurodegenerative disorder, is characterized by a variety of other motor and non-motor symptoms. As a neurotoxin, 6-OHDA can be specifically up-taken by dopaminergic neurons containing monoamine oxidase in the substantia nigra and transformed into free radical damage neurons under the action of monoamine oxidase, which has been adopted to design a cell model [14]. Our results revealed that dioscin significantly restored 6-OHDA-mediated injury in PC12 cells. As we all know, the overproduction of ROS can cause oxidative stress, resulting in the loss of DNA integrity and mitochondrial, which could damage neuronal cells [15]. Many experimental studies have confirmed that oxidative stress is closely related to neuronal apoptosis in PD and some inhibitors of the complex of mitochondrial have demonstrated the neuroprotective effects in neurodegenerative disorders [16,17]. LDH, the most sensitive enzyme to mark brain tissue damage, can catalyze the redox reaction [18]. As we all know, the overproduction of ROS can cause oxidative stress, resulting in the loss of DNA integrity and mitochondrial, which could damage neuronal cells [7]. MDA, the main metabolite of free radical damage to biological cell membrane, can cause the protein synthesis ability or synthetic protein function turbulence [19]. As an important antioxidant, GSH can participate in various redox reactions and combine with free radicals, thereby accelerating the excretion of free radicals and reducing free radical damage to important organs such as the brain [20]. The present study demonstrated that dioscin markedly alleviated LDH and ROS levels. Moreover, the levels of GSH and MDA in cells were dramatically reversed by dioscin compared with the 6-OHDA group.

Clinical symptoms of PD are mainly caused by the progressive loss of dopaminergic neurons in substantia nigra. For the PD model with unilateral injury, APO-induced rotation behavior is an important method to judge the success of the model [21]. In our study, dioscin treatment inhibited apomorphine-induced rotation in the 6-OHDA group. Previous studies have validated that the CatWalk method can be carried out for analyzing gait changes in 6-OHDA-induced animal models [22,23]. In our work, data from the study showed that dioscin decreased the time of contact between all paws and the glass plate compared with the 6-OHDA-group. In addition, the stride length which is the major symptom of PD was significantly improved after dioscin administration. However, the stand phase, step cycle, base of support and run variation were obviously increased by dioscin. Of note, these data suggested that dioscin improved motor behavior in rats. Furthermore, the results of immunohistochemical staining revealed that dioscin increased TH expression, suggesting that dioscin exerted neuroprotective effects by inhibiting dopaminergic cell loss.

Recently, it has been implicated that DUSPs play a crucial role in affecting various physiological processes [24,25]. Among them, as a cytoplasmic enzyme, DUSP6 can lead to the inactivation of ERK1/2 MAPKs [26]. In addition, a previous study has confirmed that DUSP6 can serve as an oncogene and has anti-cancer effects [27]. ERK1/2, a selective target for DUSP6, has been implicated as one of the important regulators mediating the pathogenesis and therapeutic mechanisms of bipolar disorder [28]. It has been reported that dephosphorylation of MAPKs is conducted by serine/threonine phosphatases, tyrosine phosphatases and dual-specificity MAPK phosphatases, and the inhibitor of ERK activation can completely inhibit the increase in mRNA levels of DUSP5 and DUSP6 [29]. Of note, the down-regulation of the gene DUSP6 in the prefrontal cortex mimics stress susceptibility by increasing ERK signaling and pyramidal neuron excitability in major depressive disorder [27], indicating that there may be complex regulatory relationships between the DUSP6 and ERK pathway in neurological diseases.In our work, the data showed that DUSP6 was down-regulated in vivo and in vitro, which was reversed by dioscin, and dioscin significantly decreased the expression levels of p-ERK/ERK. Although ROS may be thecause of a complex multifactorial PD, some traditional herbal medicines can prevent neurological damages caused by MPTP or 6-OHDA, such as ginsenosides and gintonin of *P. ginseng* [30]. To further investigate whether dioscin exerted a neuroprotective effect by the regulation of DUSP6/ERK-mediated oxidative stress signal, the protein levels associated with oxidative stress were detected. The results revealed that dioscin markedly deceased Keap1 expression and increased the expression levels of Nrf2, HO-1 and SOD.

Although this could be linked to the enhanced ERK activity because of the loss of DUSP6, the observation showed that the treatment of the DUSP inhibitor (E/Z)-BCI hydrochlorides can inhibit proliferation and cause apoptosis [30,31]. In our study, to explicitly evaluate the effects of dioscin on the DUSP6/ERK signal, the transfection of DUSP6 siRNA was conducted. The results displayed that cell viability was increased after pre-treatment with dioscin compared with the 6-OHDA + DUSP6 siRNA group. Similarly, the decreased protein levels of DUSP6, Nrf2, HO-1 and SOD and the increased protein levels of p-ERK and Keap1 were statistically reversed by the dioscin-treated group transfected with 6-OHDA siRNA. These findings suggested that dioscin showed the potent effect of inhibiting the Keap1/Nrf2 signal by up-regulating DUSP6, and our data placed DUSP6 as one central target in oxidative stress-induced nerve injury.

## 4. Conclusions

Collectively, we have elucidated that dioscin exerted neuroprotective effects against PD. Dioscin reversed cell viability, decreased ROS levels, improved motor behavior and TH level and restored the levels of GSH and MDA via the inhibition of oxidative stress with the involvement of the DUSP6/ERK pathway in vitro and in vivo. Therefore, our results may expand the clinical applications of the Chinese-related medicines to prevent and treat PD. Of course, further research studies are required to thoroughly elucidate the activities, mechanisms and clinical applications of this compound against PD.

## 5. Materials and Methods

### 5.1. Chemicals and Materials

Dioscin was isolated from *Dioscorea*
*nipponica* Makino in our laboratory by high- speed counter-current chromatography (HSCCC), and good separation with high purity determined by high-performance liquid chromatography (HPLC) was achieved. The compound was dissolved in dimethylsulfoxide (DMSO) for cell experiments and 0.5% sodium carboxymethyl cellulose for in vivo test [32,33]. 3-(4,5)-dimethylthiahiazo(-z-y1)- 3,5-di-phenytetrazoliumromide (MTT) was purchased from Soleb Biotech from MCE Biotech. Dulbecco’s Modified Eagle’s medium (DMEM) and fetal bovine serum (FBS) were purchased from Gibco (California, USA). ROS probe kit, protein marker and Western blot kits were purchased from TransGen Biotech (Beijing, China). The kits of glutathione (GSH), malondialdehyde (MDA) and lactate dehydrogenase (LDH) were purchased from Nanjing Jiancheng Biotechnology Company (Nanjing, China). 6-hydroxydo -pamine (6-OHDA) reagent was purchased was from Sigma Aldrich and Steraloids. Sense 5′-AAACTGTGGTGTCTTGGTACAT-3′ for DUSP6 siRNA and sense 5′-AATTCTC-CGACACGTGTCACT-3′ for the negative control siRNA was designed and obtained from GenePharma Co., Ltd. (Shanghai, China). Lipofectamine2000 reagent was purchased from Thermo Fisher Scientific (Waltham, MA, USA).

### 5.2. Cell Culture

PC12 cells were purchased from the Shanghai Institute of Biochemistry and Cell Biology (Shanghai, China). PC12 cells were cultured in DMEM medium. In total, 10% fetal bovine serum and 100 U/mL penicillin and streptomycin were added into the culture medium, which was incubated at 37°C, 5%CO_2_ and saturated humidity.

### 5.3. 6-OHDA-Induced Cell Injury

PC12 cells at the concentration of 1 × 10^5^/mL (100 μL) were seeded into a 96-well plate. After the cells were cultured for 24 h, 6-OHDA (0, 25, 50, 100, 200 and 400 μM) was administered for 6, 12, 24 and 48 h, respectively. Then, 10 μL MTT (5.0 mg/mL) was added and incubated for 4 h. After that, 150 μL DMSO was added to each well. The absorbance value at 490 nm was immediately measured with a microplate reader (Thermo Fisher Scientific, MA, USA), and cell viability was calculated. Cell morphology under ×200 magnification was photographed with a phase contrast light microscope (Nikon, Tokyo, Japan) andeach experiment was repeated 6 times.

### 5.4. Dioscin Toxicity Test

Well-cultivated PC12 cells were collected at 1200 g/min for 5 min, and the concentration of cells at 1 × 10^5^/mL were plated into 96-well plates. After 12 h, the cells were given the different concentrations of dioscin (0, 72, 145, 290, 580 and 1160 nM) under 6, 12 and 24 htreatment. Cell viabilities were conducted by MTT method as described above and a suitable concentration of dioscin was optimized [34].

### 5.5. Effect of Dioscin on 6-OHDA-Induced Cell Viability

PC12 cells were collected at 1200 g/min for 5 min. Then, the cells at 1 × 10^5^/mL (100 μL) were seeded into 96-well plates and incubated for 12 h. The cells in the treatment groups were treated with different concentrations of dioscin (145, 290 and 580 nM) for 12 h, and the cells in the model group were treated with 400 μM of 6-OHDA. MTT method asdescribed above was applied to test cell viabilities [9].

### 5.6. Detection of ROS Level

PC12 cells were collected and inoculated in 24-well plates. After treatment with dioscin or 6-OHDA as described above, 500 μL of dichlorodihydrofluorescein diacetate (DCFH-DA) was added into the plates for 1 h at 37 °C [35]. After washing, the images were obtained by fluorescent microscopy (Olympus, Tokyo, Japan) with ×200 magnification.

### 5.7. Detection of GSH and MDA Levels

PC12 cells (5 × 10^4^ cells/well) and serum of rats were collected, and the levels of GSH and MDA were detected by kits according to the manufacturer’s instruction [36].

### 5.8. Animals and Treatments

Seventy male Sprague Dawley (SD) rats weighing 180–220 g were obtained from the Experimental Animal Center at Dalian Medical University (Dalian, China) (SCXK: 2013-0006). All animals were housed at constant temperature (23 ± 3°C) and relative humidity (60 ± 10%). After two weeks of adaptive feeding, the rats were randomly divided into six groups(10 animals in each group)including the control group, 6-OHDA model group, dioscin high dose group (6-OHDA + Dio 60 mg/kg), dioscin medium dose group (6-OHDA + Dio 30 mg/kg), dioscin low dose group (6-OHDA + Dio 15 mg/kg) and sham group. Then, therats in the dioscin treatment groups were oral administrated dioscin once daily for 2 weeks. Lastly, behavioral test and gait analysis were detected. After the detection, all animals were sacrificed by the fatal dose of isoflurane, and the serum samples and brain tissues were collected. The tissues were removed and stored in −80°C.All animal care and experimental procedures were conducted in accordance with legislation regarding the Use and Care of Laboratory Animals (People’s Republic of China) and approved by the Animal Care and Use Committee of Dalian Medical University. Animal studies are reported in compliance with the guidelines [37,38].

### 5.9. 6-OHDA-Induced Injury Model in Rats

Surgical procedures were based on the previously described methods [39]. After 4% isoflurane anesthesia, the rats were placed in a stereotaxic framea and 6-OHDA (10 µg dissolved in 0.9% saline with 0.02% ascorbic acid) was injected into the right median forebrain bundle (MFB, coordinates: anteroposterior, −1.8 mm; lateral (L), 1.6 mm from bregma; ventral (V), 8.2 mm from the dura). The animals in the sham group received an equivalent volume of 0.9% saline containing 0.02% ascorbic acid as a negative control.

### 5.10. Apomorphine-Induced Rotation

Apomorphine (0.5 mg/kg in 0.1% ascorbic acid)-induced contralateral rotation was recorded using a digital video camera to count the number of rotations for 60 min at weekly intervals up to 4 weeks post-lesion using the same test conditions. The number of rotations was calculated (contralateral minus ipsilateral full turns) [40].

### 5.11. Gait Analysis by Catwalk

The gait of the rats was analyzed with CatWalk XT and more details have been described previously [41]. Rats were trained to walk across the glass walkway at least 5 times a day. Successful runs were recorded when the tracks were straight without any interruption or hesitation. Data that failed the CatWalk training were abandoned from the study. An average number of 5 replicate crossings made by each rat was recorded. At least three cycles of complete steps were analyzed crossings by CatWalk software. Rats were subjected to CatWalk from day 7 to day 14 after the administration of 6-OHDA and dioscin.

### 5.12. Histopathological and Immunochemical Examination

Brain tissues of the animals were quickly dissected and collected. The brain tissue was divided into two parts, one of which was quickly separated from the striatum on ice and stored at −80 °C. The other part was fixed in 4% paraforma-dehyde overnight at 4 °C. Then, the tissue was embedded in paraffin. The sections were stained with hematoxylin and eosin. For immunohistochemical assays, the brain was prefixed via perfusion with 4% paraformaldehyde and the tissues were sectioned using a freezing microtome. Subsequently, the sections were incubated in BSA for 1 h at a room temperature, followed by incubation overnight with tyrosine hydroxylase (TH) antibody. The samples were then treated for 1 h with secondary antibodies and observed [12].

### 5.13. Immunofluorescence Analysis

The brain tissues were fixed in 10% formalin, embeddedin paraffin and sectioned into 5 μm slices. PC12 cells in the logarithmic phase were collected and planted in 24-well plates at a density of 1 × 10^5^/mL. After 24 h of incubation, the medium in the control group was changed with serum-free DMEM, and the model group was replaced with 400 μM of 6-OHDA for 24 h. The treatment groups were given dioscin for 12 h, and then 6-OHDA was added for 24 h. After washing with PBS three times, the cells were fixed with 10% formaldehyde and tritonX100, respectively, for 20 min at room temperature. Then, the cells or tissues were blocked by 2% milk for 1 h and incubated overnight at 4 °C with DUSP6. After washing with PBS, the samples were incubated with a FITC-conjugated goat anti-rabbit IgG for 1 h and stained with DAPI (5 μg/mL) for 10 min. Lastly, the images were captured using a fluorescence microscope at ×200 magnification (Olympus, Japan) [10].

### 5.14. DUSP6 siRNA Transfection Experiments In Vitro

In vitro transfection experiment was performed on PC12 cells. DUSP6 siRNA and transfection regent-lipofectamine2000 were dissolved in Opti-MEM and equilibrated for 5 min at room temperature. Then, DUSP6-targeted siRNA was mixed gently with transfection regent for 20 min to form siRNA liposomes, and cell viability, ROS levels and mitochondrial membrane potential were detected. In addition, the expression levels of DUSP6, ERK, p-ERK, Keap1, Nrf2, HO-1 and SOD were measured after 24 h of transfection [11].

### 5.15. Western Blotting Assay

Proteins of brain tissues and cells were extracted using a protein extraction kit. The BCA protein concentration determination kit was used to determine the protein concentration, and the protein was stored at −80 °C. Then, the protein samples were loaded onto SDS-PAGE gels, and transferred onto the polyvinylidene fluoride membrane (Millipore, Burlington, MA, USA). The membrane was blocked with 5% milk for 3 h at a room temperature and incubated with primary antibodies (Appendix A, see Appendix A) overnight at 4 °C. After washing, the secondary antibody was added. The expression levels of the proteins were detected by an enhanced chemiluminescence method and normalized to GAPDH and photographed by ChemiDoc™XRS Imaging System (Bio-Rad Laboratories, Hercules, CA, USA) [8].

### 5.16. Statistical Analyses

The experimental data were performed as mean ± SD using GraphPad Prism 7.0, and a *t*-test was used to compare the two groups. One-way ANOVA was adopted to compare the mean samples among multiple groups followed by Newman–Keuls test. P test was used for pairwise comparisons, and *p* < 0.05 was considered statistically significant. 

## Figures and Tables

**Figure 1 molecules-27-03151-f001:**
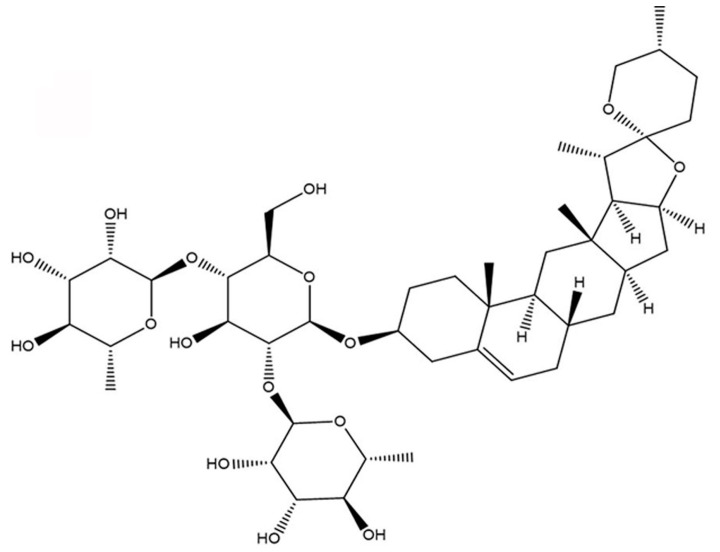
The chemical structure of dioscin.

**Figure 2 molecules-27-03151-f002:**
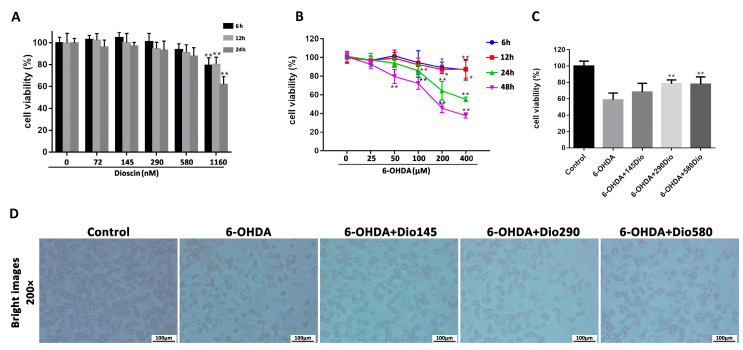
Dioscin alleviated 6-hydroxydopamine (6-OHDA)-induced injury and oxidative damage in PC-12 cells. (**A**) Cytotoxicity test of dioscin on PC12 cells. (**B**) Effect of 6-OHDA on PC12 cells. ** *p* < 0.01 and * *p* < 0.05, compared with the control group. (**C**) Protective effects of dioscin on 6-OHDA-induced cell injury. (**D**) Effects of dioscin on the morphological changes in PC12 cells. Data are expressed as mean ± SD (*n* = 6). ** *p* < 0.01 compared with the 6-OHDA group.

**Figure 3 molecules-27-03151-f003:**
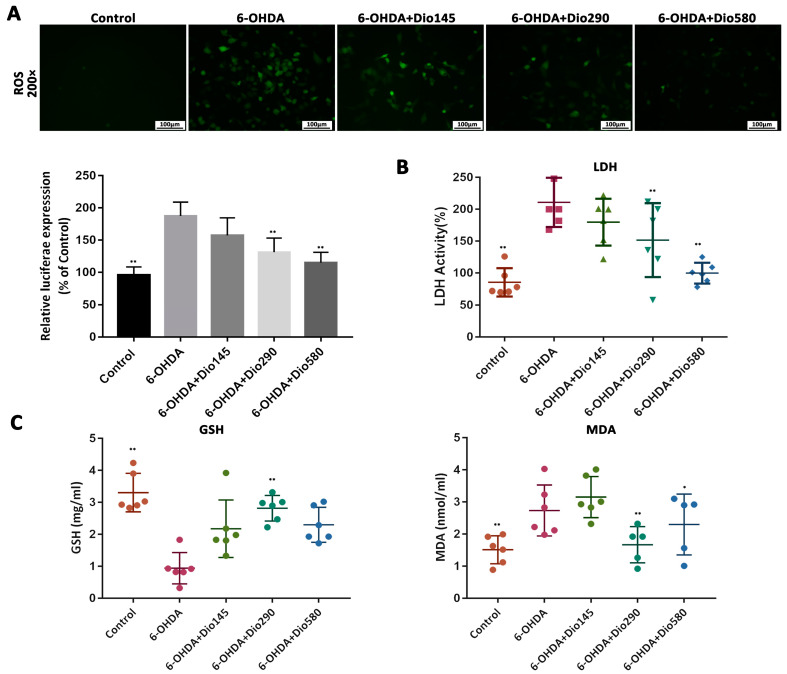
Dioscin normalized 6-hydroxydopamine (6-OHDA)-induced oxidative damage in PC12 cells. (**A**) Effects of dioscin on intracellular ROS level in PC12 cells (×200 magnification). (**B**,**C**) The levels of LDH, MDA and GSH in PC12 cells. Data are expressed as mean ± SD (*n* = 6). ** *p* < 0.01 and * *p* < 0.05 compared with the 6-OHDA group.

**Figure 4 molecules-27-03151-f004:**
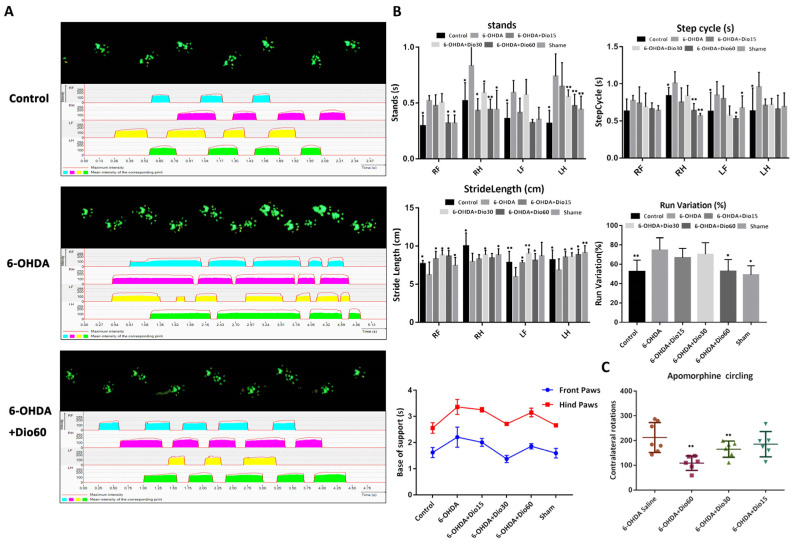
Regulation effect of dioscin gait on 6-hydroxydopamine (6-OHDA)-induced rats. (**A**) Effects of dioscin on walking pattern and individual paws (RF: right front, RH: right hind, LF: left front, and LH: left hind). The upper panel (green prints in black background) showed the digitized prints, while the lower panel (with colorful phase lags) showed the stance phase duration of each individual paw in a single step cycle. (**B**) Effects of dioscin on several parameters of CatWalk after 6-OHDA treatment. The stand phase, step cycle, stride length, run variation and base of support were tested. (**C**) Effects of dioscin on motor impairment by apomorphine-induced rotation test in rats. Data are expressed as mean ± SD (*n* = 6). * *p* < 0.05 and ** *p* < 0.01 compared with the rats in the 6-OHDA group.

**Figure 5 molecules-27-03151-f005:**
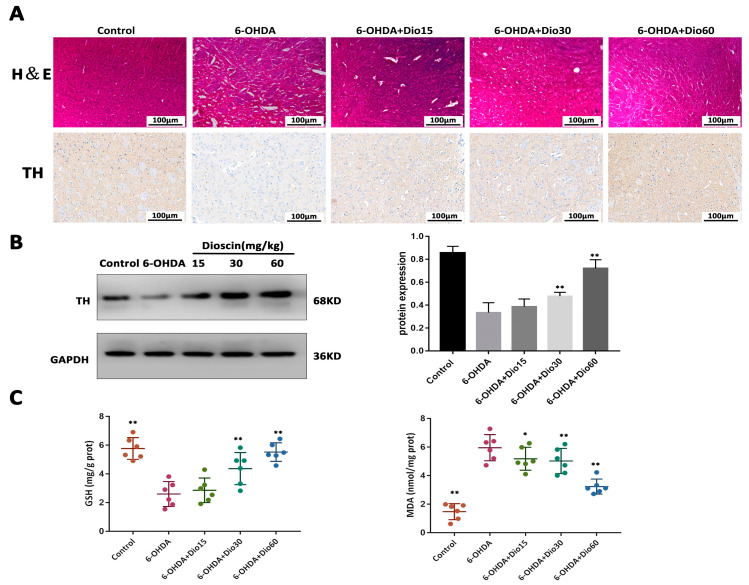
Regulation effect of dioscin on tyrosine hydroxylase (TH), glutathione (GSH) and malondialdehyde (MDA) levels. (**A**,**B**) Effects of dioscin on tissue structure and TH expression in rats (*n* = 3). (**C**) Effects of dioscin onGSH and MDA expression in rats (*n* = 6). All data are given as mean ± SD. * *p* < 0.05 and ** *p* < 0.01 compared with the 6-OHDA group.

**Figure 6 molecules-27-03151-f006:**
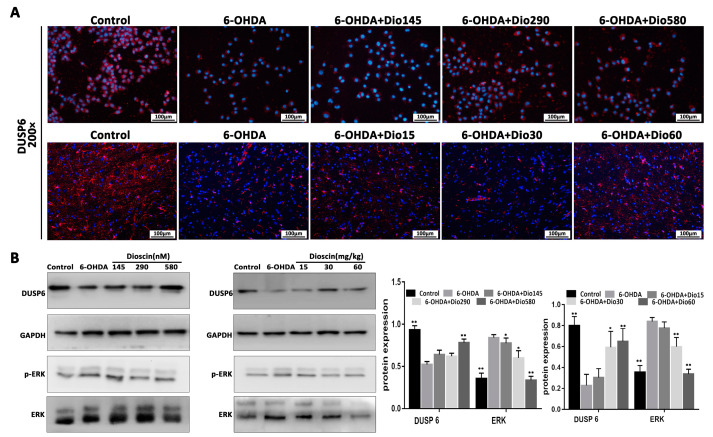
Effects of dioscin on dual-specificity phosphatase 6 (DUSP6) and phospho-extracellular regulated protein kinases (p-ERK) expression levels. (**A**) Effects of dioscin on the expression levels of DUSP6 in cells and rats by immunofluorescence assay (×200 magnification). (**B**) Western blot assays of the expression levels of DUSP6 and p-ERK/ ERK in PC12 cells and rats. Data are expressed as the mean ± SD (*n* = 3). * *p* < 0.05 and ** *p* < 0.01 compared with the 6-OHDA group.

**Figure 7 molecules-27-03151-f007:**
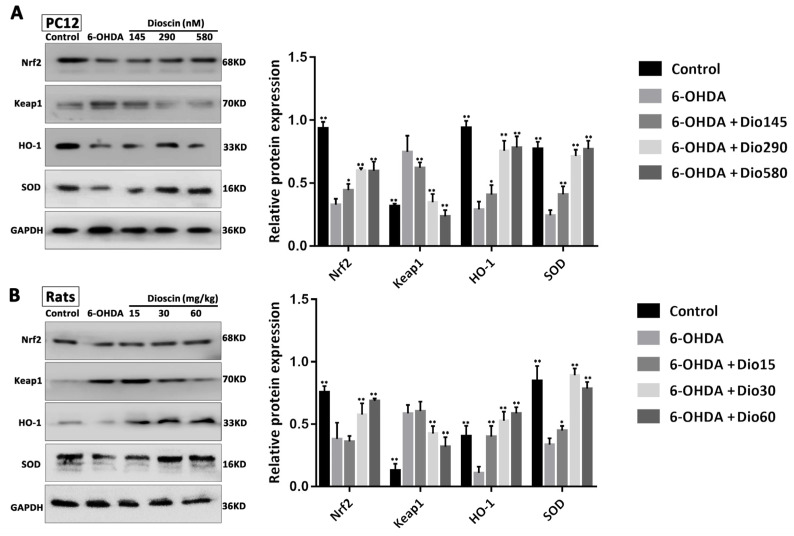
Effects of dioscin on Keap1/Nrf2 signal. (**A**) Effects of dioscin on the expression levels of Keap1, Nrf2, HO-1 and SOD by Western blotting assay in PC12 cells. (**B**) Effects of dioscinon the expression levels of Keap1, Nrf2, HO-1 and SOD by Western blotting assay in rats. Data are expressed as the mean ± SD (*n* = 3). * *p* < 0.05 and ** *p* < 0.01 compared with the 6-OHDA group.

**Figure 8 molecules-27-03151-f008:**
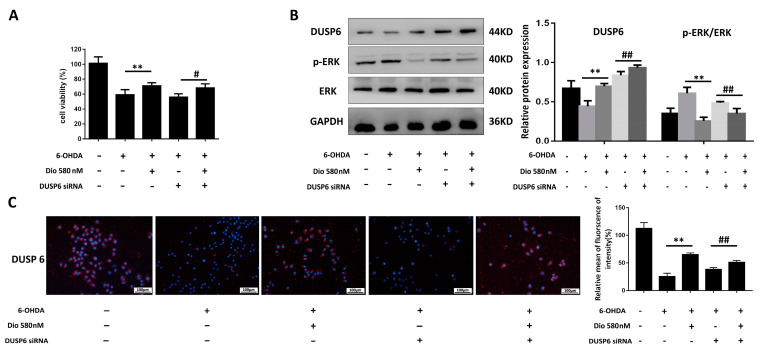
DUSP6 siRNA affected the effect of dioscin on the DUSP6 level in 6-OHDA-induced cells. (**A**) Effects of dioscin on cell viability after DUSP6 siRNA transfection (*n* = 6). (**B**) Expression levels of DUSP6, p-ERK/ERK in PC12 cells treated with dioscin after the transfection of DUSP6 siRNA (*n* = 3). (**C**) Expression level of DUSP6 in PC12 cells treated with dioscin after the transfection of DUSP6 siRNA by immunofluorescence assay (*n* = 3). Data are expressed as the mean ± SD.** *p* < 0.01 compared with the 6-OHDA group; ^#^
*p* < 0.05 and ^##^
*p* < 0.01 compared with the 6-OHDA + DUSP6 siRNA group.

**Figure 9 molecules-27-03151-f009:**
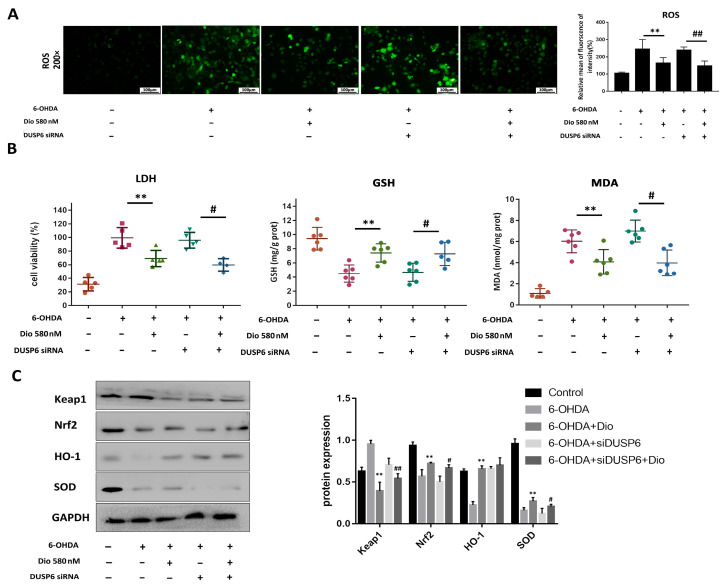
DUSP6 siRNA affected the neuroprotective effect of dioscin on the 6-OHDA-induced oxidative stress injury in PC12 cells. (**A**) Effects of dioscin on the ROS level after DUSP6 siRNA transfection (*n* = 3). (**B**) The levels of LDH, GSH and MDA in PC12 cells after DUSP6 siRNA transfection (*n* = 6). (**C**) Effects of dioscin on the expression levels of Keap1, Nrf2, HO-1 and SOD after DUSP6 siRNA transfection (*n* = 3). Data are expressed as the mean ± SD. ** *p* < 0.01 compared with the 6-OHDA group; ^#^
*p* < 0.05 and ^##^
*p* < 0.01 compared with the 6-OHDA + DUSP6 siRNA group.

## Data Availability

Not applicable.

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
