# Peer review of "Neuroprotective Effect of Dioscin against Parkinson’s Disease via Adjusting Dual-Specificity Phosphatase 6 (DUSP6)-Mediated Oxidative Stress"

_molecules, 2022, doi:10.3390/molecules27103151_

Round 1

Reviewer 1 Report

The manuscript molecules-1699692, is an interesting study about the mechanism associated with the effect of dioscin against Parkinson's disease.

Although the work is interesting and experimentally well planned, for a person who is not completely familiar with the subject, it is difficult to follow, due to the higher number of used abbreviations, some of which are not well defined in the text. So, it would be appropriate to add a glossary.

In the abstract, it is necessary to define 6-OHDA, TH, SOD and all the used abbreviations, also, the first time they are mentioned in the text. In addition, also in abstract, it is necessary to add quantitative information about the results. Likewise, all abbreviations must be defined in the figure captions.

In lines 36 and 45 the references are missing.

Please add a hypothesis at the end of the introduction.

In 2.1, please add the way to the dioscin mas obtained. Was it synthesized? Purified from plant?

In 2.3.6, please indicate the number of replicates

In 2.3-2.15, please indicate the references of the all used methodologies.

In line 89, 105, 5 must be in superscript

In the results section, these are described very precisely. Please expand the explanations in more detail.

In the discussion, LDH, ROS, GSH and MDA are discussed, without defining or giving more context of what these factors imply. Please complete.

Lines 332-335: The discussion of the involved mechanisms must be expanded.

Lines 362-365: Please, expand the  discussion of the projections of the work.

Author Response

Replaying to the comments from Reviewer 1:

The manuscript molecules-1699692, is an interesting study about the mechanism associated with the effect of dioscin against Parkinson's disease.

Although the work is interesting and experimentally well planned, for a person who is not completely familiar with the subject, it is difficult to follow, due to the higher number of used abbreviations, some of which are not well defined in the text. So, it would be appropriate to add a glossary.

In the abstract, it is necessary to define 6-OHDA, TH, SOD and all the used abbreviations, also, the first time they are mentioned in the text. In addition, also in abstract, it is necessary to add quantitative information about the results. Likewise, all abbreviations must be defined in the figure captions.

Response: Thank you for giving us the valuable comment. All the statements in abbreviationshave been defined in abstract and figure captions, and quantitative information about the results has been added into the abstract. Please see the revised abstract colored in red.

In lines 36 and 45 the references are missing.

Response: Thank you for giving us the valuable comment.The missing references have been added into the manuscript. Please see the revision on pages2 and 14.

  1. Wang, H.; Liu, D.; Sun, Y.; Meng, C.; Tan, L.; Song, C.; Qiu, X.; Liu, W.; Ding, C.; Ying, L., Upregulation of DUSP6 impairs infectious bronchitis virus replication by negatively regulating ERK pathway and promoting apoptosis. Vet Res 2021, 52, 7.
  2. Tao, X.; Yin, L.; Xu, L.; Peng, J., Dioscin: A diverse acting natural compound with therapeutic potential in metabolic diseases, cancer, inflammation and infections. Pharmacol Res 2018, 137, 259-269.

Please add a hypothesis at the end of the introduction.

Response: Thank you for your good suggestion. Thehypothesis has been added into the end of introduction. Please see the revision on page 2.

In 2.1, please add the way to the dioscinwas obtained. Was it synthesized? Purified from plant?

Response: Thank you for giving us the valuable comment.Dioscinwas isolated from Dioscorea nipponica Makino in our laboratory with the purity >98%determined by high-performance liquid chromatography (HPLC) [14,15]. Please see the revision on page 2.

  1. Yin, L.-H.; Xu, L.-N.; Wang, X.-N.; Lu, B.-N.; Liu, Y.-T.; Peng, J.-Y., An Economical Method for Isolation of Dioscin from Dioscoreanipponica Makino by HSCCC Coupled with ELSD, and a Computer-Aided UNIFAC Mathematical Model. Chromatographia 2009, 71, 15-23.
  2. Hu, M.; Xu, L.; Yin, L.; Qi, Y.; Li, H.; Xu, Y.; Han, X.; Peng, J.; Wan, X., Cytotoxicity of dioscin in human gastric carcinoma cells through death receptor and mitochon -drial pathways. J Appl Toxicol 2013, 33, (8), 712-22.

In 2.3.6, please indicate the number of replicates

Response: Thank you for your valuable comment.In6-OHDA induced cell injury, each experiment was repeated at least 6 times. The statements have been added into the manuscript. Please see the revised section colored in red.

In 2.3-2.15, please indicate the references of the all used methodologies.

Response: Thank you for giving us the valuable comment. The references of MTT assay, ROS level detection, and determination of GSH and MDA levels have been added into the sections.Please see the revision on pages3 and 14.

In line 89, 105, 5 must be in superscript

Response: Thank you for your good comment. The incorrect labeling has been revised. Please see the section colored in red.

In the results section, these are described very precisely. Please expand the explanations in more detail.

Response: Thank you. More details have been attached into the results sections based on your comment. Please see the revisions on pages5 to 10 colored in red.

In the discussion, LDH, ROS, GSH and MDA are discussed, without defining or giving more context of what these factors imply. Please complete.

Response: Thank you for giving us the valuable comment.LDH, the most sensitive enzyme to mark brain tissue damage, can catalyze the redox reaction [28].As we all known, overproduction of ROS can cause oxidative stress, resulting in the loss of DNA integrity and mitochondrial, which could damage neuronal cells. MDA is the main metabolite of free radical damage to biological cell membrane, which can cause the protein synthesis ability or synthetic protein function turbulence[29]. As an importantantioxidant, GSH participates in various redox reactions, which can combine with free radicals, thereby accelerating the excretion of free radicals andreducing free radical damage to important organs such as brain [30]. Please see the revision on pages 12 and 15.

  1. Sun, K.; Fan, J.; Han, J., Ameliorating effects of traditional Chinese medicine preparation, Chinese materia medica and active compounds on ischemia/ reperfusion-induced cerebral microcirculatory disturbances and neuron damage. Acta pharm Sin B 2015, 5, 8-24.
  2. Wang, S.; Liu, W.; Wang, J.; Bai, X., Curculigoside inhibits ferroptosis in ulcerative colitis through the induction of GPX4. Life Sci 2020, 259, 118356.
  3. Zhao, L.; Qi, Y.; Xu, L.; Tao, X.; Han, X.; Yin, L.; Peng, J., MicroRNA-140-5p aggravates doxorubicin-induced cardiotoxicity by promoting myocardial oxidative stress via targeting Nrf2 and Sirt2. Redox Biol 2018, 15, 284-296.

Lines 332-335: The discussion of the involved mechanisms must be expanded.

Response: Thank you for giving us the valuable comment.ERK1/2, a selective target for DUSP6, has been implicated as one of the important regulators mediating the pathogenesis and therapeutic mechanisms of bipolar disorder [38]. It has been reported that dephosphorylation of MAPKs is conducted by ser-ine/threonine phosphatases, tyrosine phosphatases, and dual specificity MAPK phos-phatases, and the inhibitor of ERK activationcan completely inhibit the increases on mRNA levels of DUSP5 and DUSP6 [39].Of note, downregulation of the gene DUSP6 in prefrontal cortex mimics stress susceptibility by increasing ERK signaling and pyram-idal neuron excitability in major depressive disorder [40], indicating that there may be complex regulatory relationships between DUSP6 and ERK pathways in neurological diseases. Please see the revision on pages 13 and 16.

  1. Kim, S. H.; Shin, S. Y.; Lee, K. Y.; Joo, E. J.; Song, J. Y.; Ahn, Y. M.; Lee, Y. H.; Kim, Y. S., The genetic association of DUSP6 with bipolar disorder and its effect on ERK activity. Prog Neuropsychopharmacol Biol Psychiatry 2012, 37, 41-49.
  2. Labonte, B.; Engmann, O.; Purushothaman, I.; Menard, C.; Wang, J.; Tan, C.; Scarpa, J. R.; Moy, G.; Loh, Y. E.; Cahill, M.; Lorsch, Z. S.; Hamilton, P. J.; Calipari, E. S.; Hodes, G. E.; Issler, O.; Kronman, H.; Pfau, M.; Obradovic, A. L. J.; Dong, Y.; Neve, R. L.; Russo, S.; Kazarskis, A.; Tamminga, C.; Mechawar, N.; Turecki, G.; Zhang, B.; Shen, L.; Nestler, E. J., Sex-specific transcriptional signatures in human depression. Nat Med 2017, 23, 1102-1111.
  3. Higa, T.; Takahashi, H.; Higa-Nakamine, S.; Suzuki, M.; Yamamoto, H., Up- regulation of DUSP5 and DUSP6 by gonadotropin-releasing hormone in cultured hypothalamic neurons, GT1-7 cells. Biomed Res 2018, 39, 149-158.

Lines 362-365: Please, expand the discussion of the projections of the work.

Response: Thank you for giving us the valuable comment. The discussion of the projections of the work was expanded as: “Collectively, we have elucidated that dioscin exerted the neuroprotective effects against PD. Dioscin reversed cell viability, decreased ROS levels, improved motor behavior and TH levels, and restored the levels of GSH and MDA via inhibition of oxidative with the involvement of DUSP6/ ERK pathway in vitro and in vivo. Therefore, our results may expand the clinical applications of the related-Chinese medicines to prevent and treat PD. Of course, further researches are required to thoroughly elucidate the activities, mechanisms and clinical applications of this compound against PD”. Please see the revision colored in red.

Reviewer 2 Report

Dear Authors,

The manuscript presents a theme of significant scientific relevance. However, I have some comments.
 1. In the Material and Methods,
1.1 In the animals and treatments, what was the time for each treatment?
1.2 How many animals were used in each treatment?

2. In the Results,

2.1 Why did you use 3 animals to evaluate the effects of dioscin on tissue structure and 6 animals for oxidative stress parameters? (Figure 4)
2.2 Western-Blotting results are unclear. Is it possible to attach the stained gels and membranes from the experiments?

Author Response

Replaying to the comments from Reviewer 2:

The manuscript presents a theme of significant scientific relevance. However, I have some comments.
 1. In the Material and Methods,
1.1 In the animals and treatments, what was the time for each treatment?

Response: Thank you for your valuable comment. The rats in dioscin treatment groups were oral administrated with dioscin once daily for 2 weeks.

1.2 How many animals were used in each treatment?

Response: Thank you for giving us the valuable comment. Ten rats were used in each group for the tests. The statement has been attached into the section of animals and treatments colored in red.

  1. In the Results,

2.1 Why did you use 3 animals to evaluate the effects of dioscin on tissue structure and 6 animals for oxidative stress parameters? (Figure 4)

Response:Thank you for giving us the valuable comment. In our study, the number三 of animal experiments design are based on the content and general principles of the experiment. Whether the numbers of samples are 3 or 6, we guaranteed the principle of random selection and repetition of samples. For kits tests, six samples were usually randomly used for biological repetition. For other experiments, three samples were used from the perspective of funds, as well as saving samples for other different experiments, which are consistent with previous references [1-4].

  1. Gao, Q.; Gu, Y.; Jiang, Y.; Fan, L.; Wei, Z.; Jin, H.; Yang, X.; Wang, L.; Li, X.; Tai, S.; Yang, B.; Liu, Y., Long non-coding RNA Gm2199 rescues liver injury and promotes hepatocyte proliferation through the upregulation of ERK1/2. Cell Death Dis 2018, 9, 602.
  2. Wang, L.; Yang, H.; Wang, Q.; Zhang, Q.; Wang, Z.; Zhang, Q.; Wu, S.; Li, H., Paraquat and MPTP induce alteration in the expression profile of long noncoding RNAs in the substantia nigra of mice: Role of the transcription factor Nrf2. Toxicol Lett 2018, 291, 11-28.
  3. Lin, C. M.; Lin, Y. T.; Lin, R. D.; Huang, W. J.; Lee, M. H., Neurocytoprotective Effects of Aliphatic Hydroxamates from Lovastatin, a Secondary Metabolite from Monascus-Fermented Red Mold Rice, in 6-Hydroxydopamine (6-OHDA)-Treated Nerve Growth Factor (NGF)-Differentiated PC12 Cells. ACS Chem Neurosci 2015, 6, 716-24.
  4. Law, V.; Dong, S.; Rosales, J. L.; Jeong, M. Y.; Zochodne, D.; Lee, K. Y., Enhancem -ent of Peripheral Nerve Regrowth by the Purine Nucleoside Analog and Cell Cycle Inhibitor, Roscovitine. Front Cell Neurosci 2016, 10, 238.

2.2 Western-Blotting results are unclear. Is it possible to attach the stained gels and membranes from the experiments?

Response: Thank you for giving us the valuable comment.In order to make the results of protein bands clearer, we replaced some of poor qualitybands, includingHO-1 of rats in Fig. 6, HO-1 in Fig. 8, and ERK in Fig. 7.Please see the revised Fig.6 , Fig. 7 and Fig. 8.

Round 2

Reviewer 1 Report

The manuscript molecules-1699692 has been highly improved with respect to its original version, incorporating practically all the comments.

Some minor comments:

The phrase incorporated at the end of the introduction does not correspond to a hypothesis, please rewrite.

Line 65: How was Dioscin isolated? By HPLC-prep? HSCCC?. Please complete the information.

2.1: MTT, GSH, MDA, LDH and 6-OHDA must be defined (first mentioned in text)

Line 67: Please, define DMSO.

Please, add references in 2.5, 2.12-2.15.

Line 106: Please replace doscin by dioscin.

Line 117: (10 animals in each group)including control group. The space is missing between group) and including.

Line 150: replace werequickly by were quickly

Author Response

Replaying to the comments from Reviewer 1:

The manuscript molecules-1699692 has been highly improved with respect to its original version, incorporating practically all the comments.

Some minor comments:

The phrase incorporated at the end of the introduction does not correspond to a hypothesis, please rewrite.

Response: Thank you for giving us the valuable comment. The hypothesis at the end of the introduction has been re-written. Please see the revision colored in red on page 2.

Line 65: How was Dioscin isolated? By HPLC-prep? HSCCC?. Please complete the information.

Response: Thank you for your valuable comment. Dioscin was isolated from D. nipponica Makino by high-speed counter-current chromatography (HSCCC), and good separation was obtained. The information has been attached into the section. Please see the revision colored in red on page 2.

2.1: MTT, GSH, MDA, LDH and 6-OHDA must be defined (first mentioned in text)

Line 67: Please, define DMSO.

Response: Thank you for giving us the valuable comment. The MTT, GSH, MDA, LDH, 6-OHDA and DMSO have been defined. Please see the revision colored in red on page 2.

Please, add references in 2.5, 2.12-2.15.

Response: Thank you for your comment. The references have been added into the section. Please see the revision colored in red on pages 3 to 5.

Line 106: Please replace doscin by dioscin.

Response: The spelling mistake has been corrected in line 106. Thanks.

Line 117: (10 animals in each group)including control group. The space is missing between group) and including.

Response: Thank you. The spelling mistake has been corrected. Please see the revision colored in red.

Line 150: replace werequickly by were quickly

Response: Thank you for giving us the valuable comment. The spelling mistake has been corrected.